# Identifying potential sites for rainwater harvesting ponds (e*mbung*) in Indonesia's semi-arid region using GIS-based MCA techniques and satellite rainfall data

**Yulius Patrisius Kau Suni** [1,2]*, **Joko Sujono**[1]*, **Istiarto**[1]

**1** Department of Civil and Environmental Engineering, Faculty of Engineering, Universitas Gadjah Mada, Yogyakarta, Indonesia, **2** Department of Civil Engineering, Faculty of Engineering, Universitas Katolik Widya Mandira, Kupang, Indonesia

* yulius.suni@mail.ugm.ac.id (YPKS); jsujono@ugm.ac.id (JS)

**Data Availability Statement:** The data on rainfall satellite and raster format maps were used for this

## Abstract

People have used rainwater harvesting (RWH) technology for generations to a considerable extent in semi-arid and arid regions. In addition to meeting domestic needs, this technology can be utilized for agricultural purposes as well as soil and water conservation measures. Modeling the identification of the appropriate pond's location therefore becomes crucial. This study employs a Geo Information System (GIS) based multi-criteria analysis (MCA) approach and satellite rainfall data, Global Satellite Mapping of Precipitation (GSMaP) to determine the suitable locations for the ponds in a semi-arid area of Indonesia, Liliba watershed, Timor. The criteria for determining the location of the reservoir refer to the FAO and Indonesia's small ponds guideline. The watershed's biophysical characteristics and the socioeconomic situation were taken into consideration when selecting the site. According our statistical analysis, the correlation coefficient results of satellite daily precipitation were weak and moderate, but the results were strong and extremely strong for longer time scales (monthly). Our analysis shows that about 13% of the entire stream system is not suitable for ponds, whereas areas that are both good suitability and excellent suitability for ponds make up 24% and 3% of the total stream system. 61% of the locations are partially suited. The results are then verified against simple field observations. Our analysis suggests that there are 13 locations suitable for pond construction. The combination of geospatial data, GIS, a multi-criteria analysis, and a field survey proved effective for the RWH site selection in a semi-arid region with limited data, especially on the first and second order streams.

## Introduction

One technology used to lessen the likelihood of a drought is rainwater harvesting (RWH) [1–6]. RWH is a technique for gathering and preserving surface runoff in small (local) catchments

study. These data have been submitted in this revision under Supporting information.

**Funding:** The authors received no specific funding for this work.

**Competing interests:** The authors have declared that no competing interests exist.

for agricultural use [7]. Since 4000 years ago, this technology has been used, particularly in the Middle East and the Mediterranean region [8]. Evidence of RWH technology is more than 9000 years old in Jordan [9]. Farm ponds (*embung* in Indonesia) are a common form of RWH technology in Indonesia's semi-arid region. In the early 1980s, the government started constructing ponds in Timor island with assistance from international donors [10]. Prior to that, however, several ponds had already been constructed in a cattle grazing area on the north coast of the North Central Timor also known as *Timor Tengah Utara* (TTU) district in the 1970s by Timor Livestock Company (Timlico).

This water harvesting method seeks to enhance productivity by, among other things, reducing negative effects of the agricultural dry spell [11]. A study evaluating the effectiveness of several types and sizes of agricultural ponds used for supplemental irrigation and recharging open wells in India has suggested that small ponds prolonged the growing season, improve agricultural output, and boost farmers' revenue [12]. The effectiveness of water harvesting method has also been reported in the findings of a study on the impact of *jessour* (a kind of RWH, a sedimentary basin, and a planting place in Tunisia) on soil moisture was conducted in the Dahar plateau in southeast Tunisia. The findings demonstrate that the presence of *jessour* preserved the soil moist during the summer, which supported the growth of plants, particularly olive trees, in the *jessour* storage area [13]. Another study evaluating the effectiveness of technologies for artificial recharge and rainwater harvesting in Iran suggested that an artificial groundwater recharge system could inject new groundwater in volumes up to hundreds of thousands of $m^3$ during the dry season and 4.5 x 106 $m^3$ during the rainy season [8]. Similar studies have been carried out in Timor, Indonesia suggesting the effectiveness of water harvesting method, also known as *embung*, for many years [10, 14–16].

However, some studies have reported the risk of reservoir capacity silting up due to sedimentation, a risk factor that must be taken into account when designing the *embung* [10, 15, 16]. Therefore, proper site selection and appropriate engineering planning are crucial for RWH success [3]. The most popular techniques for determining the location and appropriate RWH technology, particularly in small catchment areas, are field surveys based on biophysical and socioeconomic parameters [9]. Numerous studies conducted after 2000 combined the use of hydrological modeling, Hydrologic Engineering Center-The Hydrologic Modeling System (HEC-HMS), Geographic Information System (GIS) tools, and multi-criteria analysis (MCA) [4, 6, 17–19].

Despite the previous use of these techniques reported in the aforementioned studies, none of them focused on lesser stream orders (1st and 2nd stream orders) that are suited for small reservoirs (*embungs*). Thus, this study aims to fill this gap in knowledge by developing a model for identifying potential embung locations in 1st and 2nd stream orders based on satellite data (rainfall, topography, soil conditions, and land use) using a combination of GIS-based MCA techniques.

As data on rainfall is one of the crucial factors in establishing the sites or locations of RWH, this study used Global Precipitation Measurement (GPM), product [20], a substitute for the Tropical Rainfall Measuring Mission (TRMM) satellite since 2015 [21], due to limited availability of in-situ observed rainfall data in all parts of the watershed. The majority of researchers analyze runoff depth using recorded rainfall data [6, 9, 12, 17, 19, 20, 22, 23]. A small number of studies employ satellite-based rainfall data including TRMM [24]. In Indonesia, the TRMM satellite based rainfall data has a pattern that corresponds to the observed rainfall [25]. Using GPM satellite data in semi-arid regions and small watersheds, this research seeks to address the shortfalls in in-situ observed rainfall data.

## Materials and methods

### Study area

The study area is Liliba watershed, located in Timor, East Nusa Tenggara (NTT) province of Indonesia between longitudes 123˚37'23.3"– 123˚40'32" E, and latitudes 10˚08'40.3"– 10˚15'22.6" S, (see Fig 1). This watershed is situated in Kupang City and Kupang district and covers a total area of 33.6 km². The climate data for the Liliba watershed apply to Kupang City's climate. Climate uncertainty has become a challenge in this region. The dry season lasts longer than the wet season on average per year. According to Kupang climate data [26], the lowest average air temperature between 2013 and 2020 was 21.5˚C in August, and the highest average temperature was 34.7˚C in September. The lowest and maximum monthly rainfall averages during the rainy season from 2013 through 2020 were 190 and 445 millimeters, respectively. The January period always has the most days with rain, from around 19 to 28. Fig 2 shows the distribution of the study locations' average annual rainfall based on GSMaP data.

The Indonesian National Disaster Management Agency reported that hydrometeorological disasters frequently occur in this area [28]. The most common natural catastrophes are cyclones and strong winds (43 events), followed by landslides (11), floods (10), abrasion, and drought (2 occurrences each). *Embung* construction is an option that, in turn, could mitigate droughts and floods in watershed areas.

### Data collection

The development of the spatial database is a crucial phase in most GIS operations. Digital Elevation Model (DEM), detail map of Indonesia, soil data, satellite-based meteorological data, and field observation were the datasets employed in this study. All maps were downloaded for free from the opensource websites of the FAO and the Indonesian Geospatial Information Agency (BIG). The description of these data and their sources are shown in Table 1. Each dataset was processed using ArcGIS 10.5 software. All the criteria layers were georeferenced using zone 51 S, WGS 1984, and the Universal Transverse Mercator system.

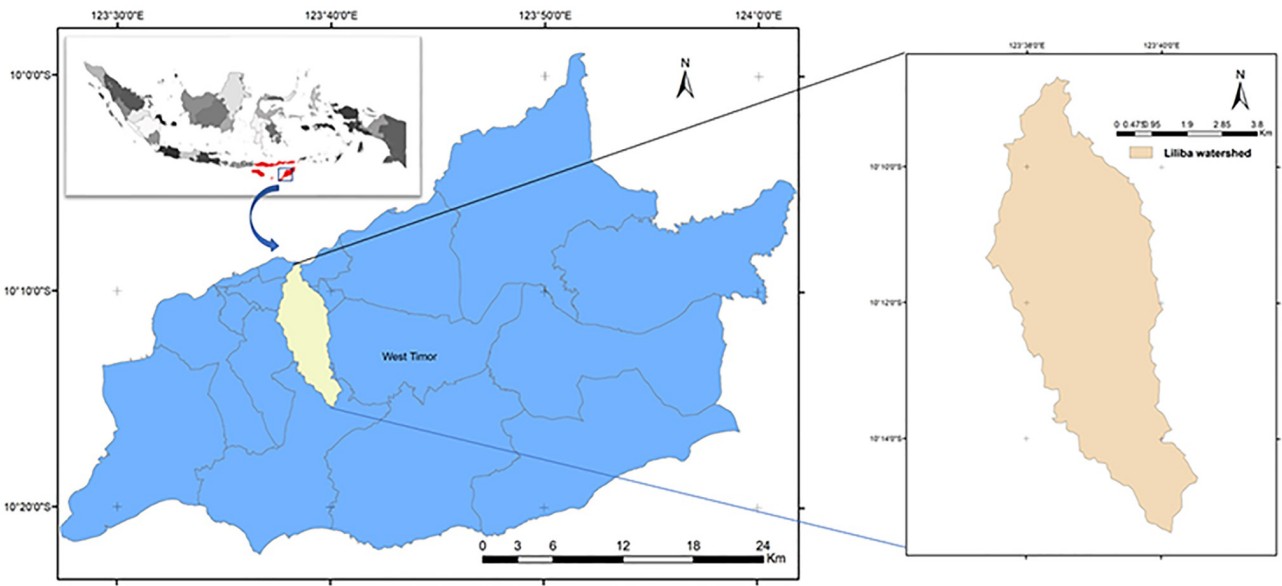

**Fig 1. The study area, Liliba watershed.** The map was derived from BIG [27].

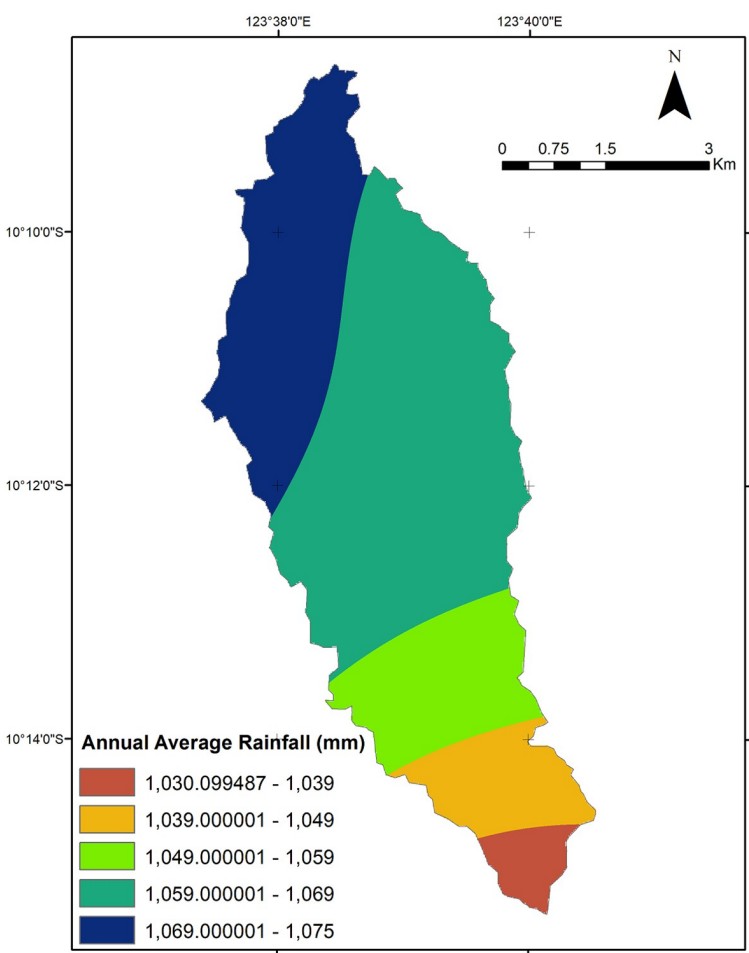

**Fig 2. Annual average rainfall distribution map.** The map was derived from BIG and GSMaP data [27, 29].

**DEM.** The DEMNAS Satu Data Indonesia website, Ina-Geoportal [27], was used to obtain DEM data (see Fig 3). This information is useful for creating flow direction, flow accumulation, river order, slope, topography, and catchment area. The DEM used is created from IFSAR 5m, TERRASAR-X 5m, and ALOS PALSAR 11.25m. Resolution is 0.27 arcseconds.

The study used DEM data that provide slopes, where slopes under 5% meet the criteria for the reservoir's site. Additionally, the study used DEM data that produce the river's order. As potential reservoir locations, the first and second orders are considered suitable.

**Land cover and land use.** In this study, land use and land cover are based on official and opensource detail map of Indonesia 2019 [27], The map was developed by Indonesia Geospatial Information authority (BIG). Based on USGS Land Use and Land Cover [30], the study area was divided into seven different categories of land use and cover, including water, built-up land, barren land, forest land, shrubland, herbaceous natural, and cultivated land, (see Fig 4). The land cover categories are statistically represented in Table 2.

While infiltration and evapotranspiration are dependent on land use patterns, the tree cover (forest), shrubland, grassland, farming, water bodies, and mangroves are a more favourable type of land cover for rainwater harvesting. In light of previous studies, the suitability of each type of land use and land cover is assessed [20, 22]. Due to socioeconomic and

**Table 1. Data type, source, year and description.**

| No | Data Type | Source | Year | Description |
|----|-----------|--------|------|-------------|
| 1 | Digital Elevation Model (DEM) | Indonesian Public Authority for Geospatial Information (BIG) (https://tanahair.indonesia.go.id/demnas/#/) License letter from BIG and its translation are attached in S2 and S3 Files. | 2012 | DEM (DEMNAS) is an opensource data created from IFSAR 5m, TERRASAR-X 5m, and ALOS PALSAR 11.25m. Resolution is 0.27 arcseconds. |
| 2 | Detail Map of Indonesia (RBI) | BIG (https://tanahair.indonesia.go.id/portal-web/download/perwilayah) License letter from BIG and its translation are attached in S2 and S3 Files. | 2019 | Opensource data. Covering information on land use and land cover (LULC) |
| 3 | Soil type | FAO | 2003 | |
| 4 | Meteorological data | Global Precipitation Measurement (GPM) Raw data of daily rainfall data from GSMaP is attached in S1 Data. | 2004–2021 | Daily rainfall data |
| 5 | Socio-economic data | Field survey | 2022 | Data on livelihood, water source for domestic use, landownership |

environmental issues, built up areas and areas with little vegetation are not suited as reservoirs. List of land use type and its suitability are summarized in Table 2.

**Soil type.** Soil map was derived from FAO website [31]. The soil map of the study area displays four different types of soil: rendzina, lithosols, dystric cambisols, and eutric cambisols.

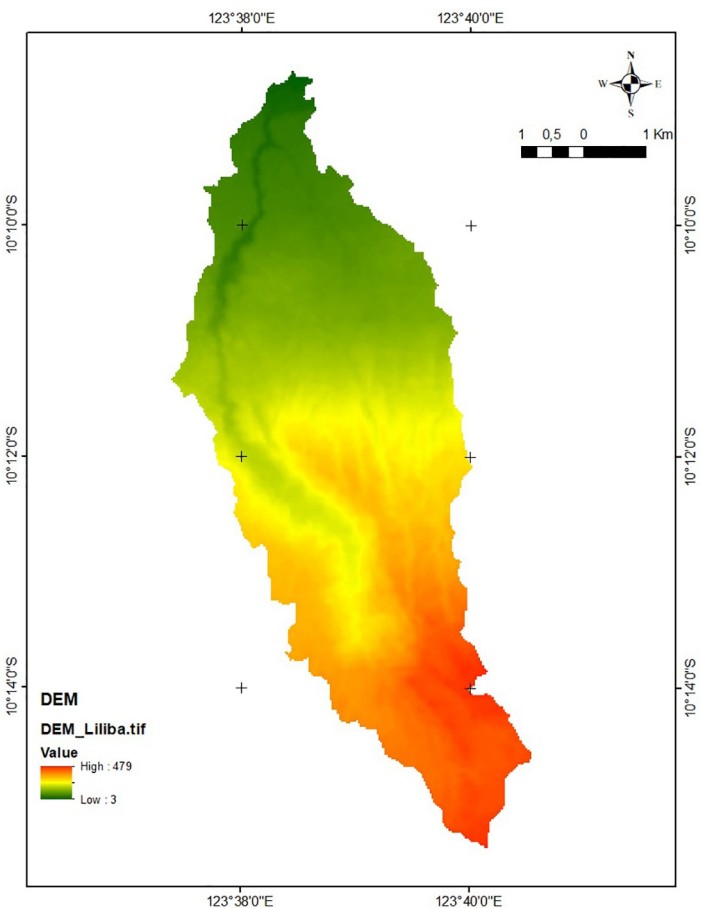

**Fig 3. The digital elevation model (DEM) of Liliba watershed.** The map was derived from BIG [27].

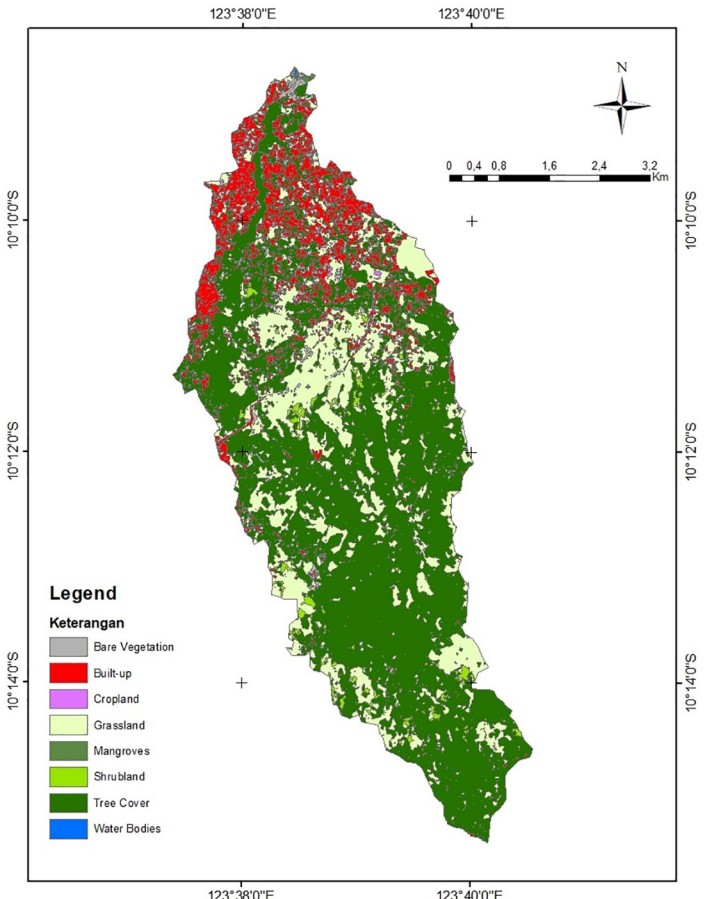

**Fig 4. Land cover and land use of study area.** The map was derived from BIG [27].

Soil type suitability was developed based on FAO standard [31]. Except for 33% of the land, which is moderately suitable since the soil is clay loam, there is no soil type in this watershed that is particularly favourable for the location of the reservoir. Sand and clay loam makes up to 42.64% of the watershed's area, with the remainder being loam. Table 3 summarizes soil type and its suitability.

**Satellite-based climate data and flow discharge.** Global Satellite Mapping of Precipitation (GSMaP) data [29] were utilized in this research due to its better performance in daily,

**Table 2. Land cover and land use statistical data.**

| Land use classes | Area km² | Area % | Suitability [20, 22] |
|---|---|---|---|
| Water | 0.005 | 0.02 | Suitable |
| Built-up | 3.971 | 11.82 | Not suitable |
| Barren | 0.850 | 2.53 | Not suitable |
| Forested land | 20.469 | 60.94 | Suitable |
| Shrubland | 0.549 | 1.63 | Suitable |
| Herbaceous natural | 7.718 | 22.97 | Not suitable |
| Cultivated land | 0.031 | 0.09 | Suitable |
| **Total** | **33.592** | **100** | |

**Table 3. Soil types and suitability in Liliba watershed.**

| Soil Group | Area % | Category [31] | Suitability [20] |
|---|---|---|---|
| Rendzinas | 9.60 | Loam | Not suitable |
| Eutric Cambisols | 14.69 | Loam | Not suitable |
| Dystric Cambisols | 33.06 | Clay loam | Moderately suitable |
| Lithosols | 42.64 | Sandy clay loam | Not suitable |

monthly and seasonal scales other than other satellite rainfall data [21]. In Indonesia, GSMaP performs better in detecting daily rainfall [32]. GSMaP is a product of GPM, the successor of TRMM JAXA satellite data [21, 33]. These data's spatial resolution is one hour, with a horizontal resolution of $0.1^0$ latitude/longitude [33]. The satellite data were evaluated using daily precipitation data from Eltari meteorological station [34]. There were no data for 2013 in the observed data search. In order to keep the analysis of satellite data consistent with observed data, data for 2013 were disregarded.

The satellite data from 2004 to 2021 were the rainfall data utilized in the hydrological analysis. Data from the Eltari station were used as the reference data. Using the Rescaled Adjusted Partial Sums (RAPS) approach, the accuracy of the rain gauge data at the Eltari station were evaluated. According to the test results. the measurement data were consistent to serve as a reference.

The Soil Conservation Service (SCS) and curve number (CN) [35] were used to estimate runoff depth values in the research area. Rainfall, hydrological soil groups, land use/cover, and soil types are the main variables that affect rainfall-runoff according to Maidment (1992) as mentioned by [19, 36]. The equations are:

$$Q = \frac{(P - 0.2\,S)^2}{P + 0.8\,S} \tag{1}$$

$$S = \frac{25400}{CN} - 254 \tag{2}$$

Where,
Q: runoff depth (mm),
P: rainfall depth (mm),
S: potential maximum retention after runoff begins (mm), and
CN: dimensionless runoff index defined based on hydrological soil group and land use.

## Multi-criteria analysis

The criteria for determining the location of the reservoir refer to the FAO and Indonesia's small ponds guideline. Runoff potential, slope, land use, soil texture, flow discharge [9, 20, 37] are among the biophysical criteria. This study introduces novel inputs for the criterion for locating the reservoir in the first and second order of streams. Location determination must take into account socio-economic issues proposed by Oweis et al (1998) and FAO (2003) as mentioned by [9] in addition to biophysical ones. The issued covers land ownership, population density, work force, people's priority, experiences with RWH, water laws, accessibility and related cost [9]. The socioeconomic factors in this study, such as land tenure [9, 18, 19], and people's preference for using pond water for domestic, agricultural, and livestock purposes [37], were selected in accordance with the context of Indonesia.

Three approaches were used to conduct multi-criteria analysis: the binary technique proposed by Eastman (1999) as cited by [38], the weighted linear combination (WLC) [39, 40] and the analytical hierarchy procedure (AHP) proposed first by Saaty (1980) as mentioned by [41]. The binary technique was used to determine whether or not the requirements for the reservoir's placement are achievable. Locations were given a score of 1 if they satisfy the assessment criteria. while those that don't were given a number of 0 [38]. The 9 criteria that this study established are listed in Table 4 below.

Following the binary method of assessment based on nine feasibility assessment criteria, the level of location suitability of various available locations was assessed using the Analytical Hierarchy Process (AHP) and Weighted Linear Combination (WLC) methods. While WLC was used to calculate the suitability value for the reservoir location AHP was utilized to estimate the weight and value of each evaluation criterion. Eq 3 determines the level of site suitability for the reservoir's placement.

$$S_i = \sum_{j}^{n} w_j . x_j \qquad (3)$$

where:

**Table 4. Criteria for embung location selection.**

| No | Criteria | Value |
|---|---|---|
| | **Biophysical aspects** | |
| 1 | The input stream's slope is less than 5%. | |
| | • < 5% | 1 |
| | • > 5% | 0 |
| 2 | Stream order | |
| | • Order 1 and 2 | 1 |
| | • ≥ Order 3 | 0 |
| 3 | Discharge of the input stream | |
| | • > 5 liter/second | 1 |
| | • < 5 liter/second | 0 |
| 4 | Soil type | |
| | • Clay | 1 |
| | • Non clay soil | 0 |
| 5 | Land cover and land use | |
| | • Forest. shrubland. grassland. cropland | 1 |
| | • Built-up. bare vegetation. water bodies. mangrove | 0 |
| | **Socio-economic aspects** | |
| 6 | Available agriculture plots. | |
| | • Available | 1 |
| | • Not available | 0 |
| 7 | The possibility of breeding livestock that can use the reservoir's water. | |
| | • Available | 1 |
| | • Not available | 0 |
| 8 | Water from reservoirs can be used for household purposes. | |
| | • Yes | 1 |
| | • No | 0 |
| 9 | Land ownership of embung site | |
| | • Clear and allowed for embung | 1 |
| | • Not clear | 0 |

Source: [1, 9, 20, 37]

$S_i$: suitable site in location i
$w_j$: weighting of factor j
$x_j$: the membership value of criteria j
$n_j$: number of parameter

Map-based processing was used to process the nine criteria. All maps were formatted into a raster file (tif) format. The raster files are attached in S1 File. Additionally, the nine criteria were applied to an overlay (multi-criteria analysis) method. The normalization technique is used to weight each criterion, with each criterion given the same weight [41]. The results of the overlay analysis were divided into five categories: unsuited (0–0.20), poor suitability (0.21–0.40), moderate suitability (0.41–0.60), good suitability (0.61–0.80), and exceptional suitability (0.81–1.00).

## Field observation

56 sub-watersheds identified by GIS spatial analysis were the focus of field observations conducted in October 2022. Downstream parts (outlets) of the 1st and 2nd order rivers are where the prospective reservoir will be located. A map of the location of the potential reservoir according to desktop analysis can be seen in Fig 5.

The use of field observations entailed examining the presence of natural valleys, conducting open interviews with the community regarding the usage of water for domestic use, livestock,

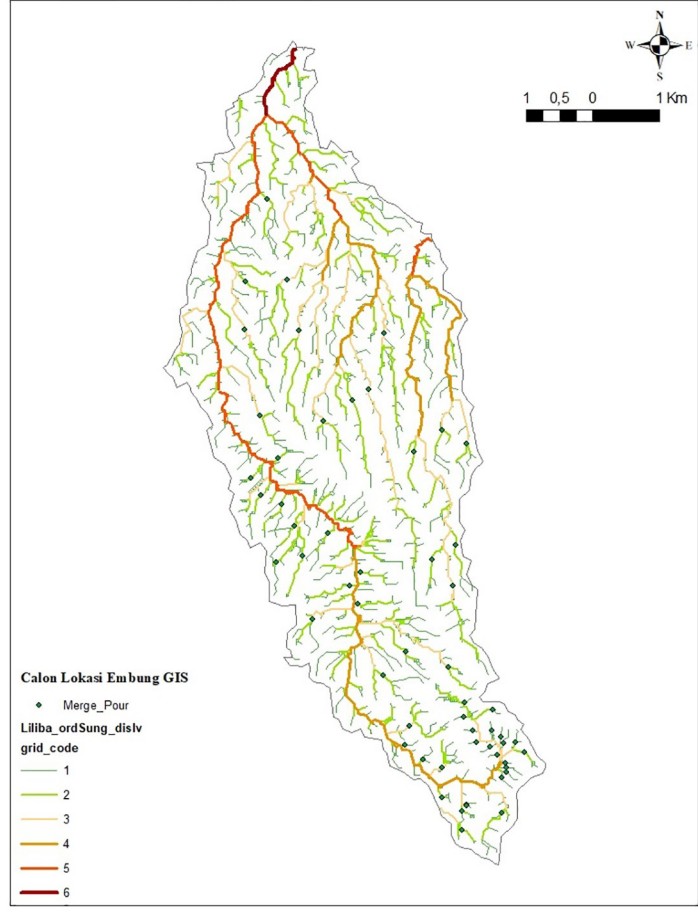

**Fig 5. Desktop based analysis of prospective sites of *embung*.**

agriculture, and determining the status of land ownership. The basin availability variable in this study served as a reference for confirming the applicability of the GIS analysis and field circumstances despite not being part of the GIS analysis itself. Fig 6 displays field observations indicating suitability for potential RWH sites.

## Results

### Satellite rainfall data analysis

**Data validation.** The correlation coefficient (r), which is the square root of the coefficient of determination ($R^2$), is 0.34 for dry season, according to statistical analysis. This demonstrates how poorly correlated the daily rainfall is (see Fig 6(a) and 6(b)). The same analysis for the rainy season shows a moderate correlation with a correlation coefficient of 0.51. The correlation analysis of monthly data shows high (r = 0.73) and very high correlation (r = 0.82) for rainy season and dry season respectively (see Fig 7(c) and 7(d)). Therefore, monthly data and a longer time scale can be used for further analysis, such as discharge analysis.

**Frequency and discharge analysis.** The application AProb 4.1 created by Istiarto [42] was used to conduct frequency analysis. The satellite data utilized for the frequency analysis is divided into two grids, the upstream grid, and the downstream grid. Since two grids of satellite rainfall data cover the study areas, two grids were chosen. Table 5 displays maximum monthly rainfall.

Flow discharge was calculated based on rainfall, soil hydrological group, and land cover. Maximum daily rainfall on the study area was 143 mm. Hydrological soil groups were B (24.3%), C (42.6%), and D (33.1%). Composite CN was calculated using information on hydrological soil groups and land cover. The CN composite was 78. The effective surface run-off flow rate (discharge) was calculated, and the average value was 50 liters/second, with the lowest of 3 liters/second and the highest at 283 liters/second. As a result, except one sub water-shed, the discharges from each sub-watershed meet the criteria for input flows (> 5 litters/second).

### Suitable sites for rainwater harvesting

Nine criteria are taken into consideration when applying an overlay for the GIS-based MCA approach to determine the potential site of the reservoir (layers). The nine variables include slope (< 5%), land use, agricultural area to be irrigated, potential for livestock to use water, domestic purposes, first and second stream order, input flow rate above 5 litters/second, soil type (clay), and the ownership status of the land where the reservoir will be built. The colour of the map indicates how suitable a site is. Red colour denotes an unsuitable site, yellow indicates a conditional location, and green represents an ideal site. The Fig 8 displays the results of the overlaying analysis performed using the MCA approach.

The analysis' findings (see Fig 9) based on biophysical aspects show that half (50%) of all stream orders of 1 and 2 meet the criteria for suitable ponds sites (good and excellent suitability). Approximately 9% of locations are unsuitable, 13% are poorly suitable, and up to 29% are moderately suitable. The number of viable places (good and excellent) drops to 27% when socioeconomic factors are considered in the analysis. Locations that are unsuitable have climbed to 13%, poorly suitable have increased significantly to 44%, and moderately suitable have declined to 17%.

The criteria for reservoir locations include areas with stream order 1 to 2 and natural valleys. The observations demonstrate that there is no natural flow identified for stream order 1 with a small catchment area. In Kupang City's, various examples like this occur. Land use changes use, particularly from agricultural land to settlements, are one of the causes. The

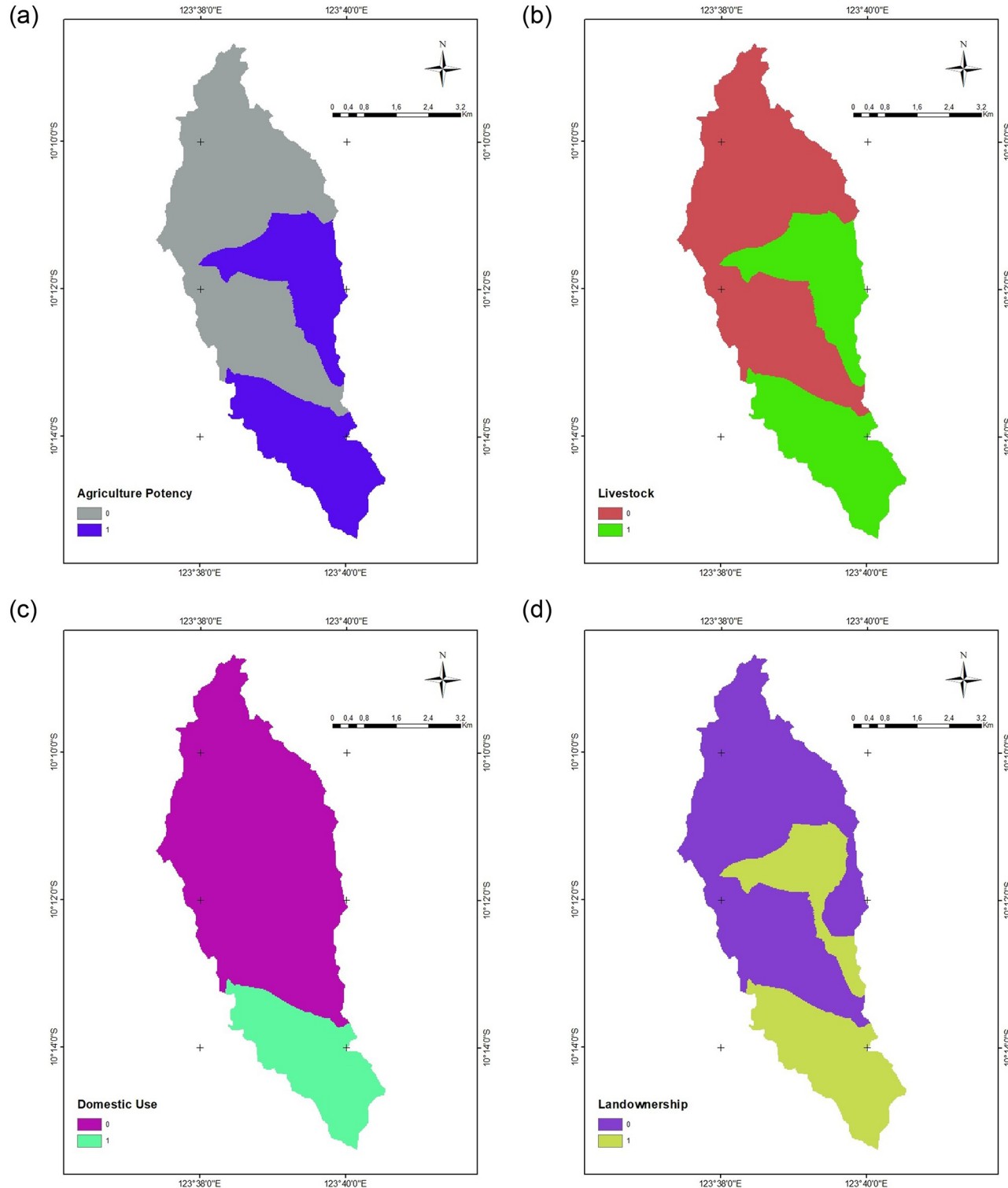

**Fig 6. Field observation, (a) agriculture potency, (b) livestock, (c) domestic use, (d) landownership.**

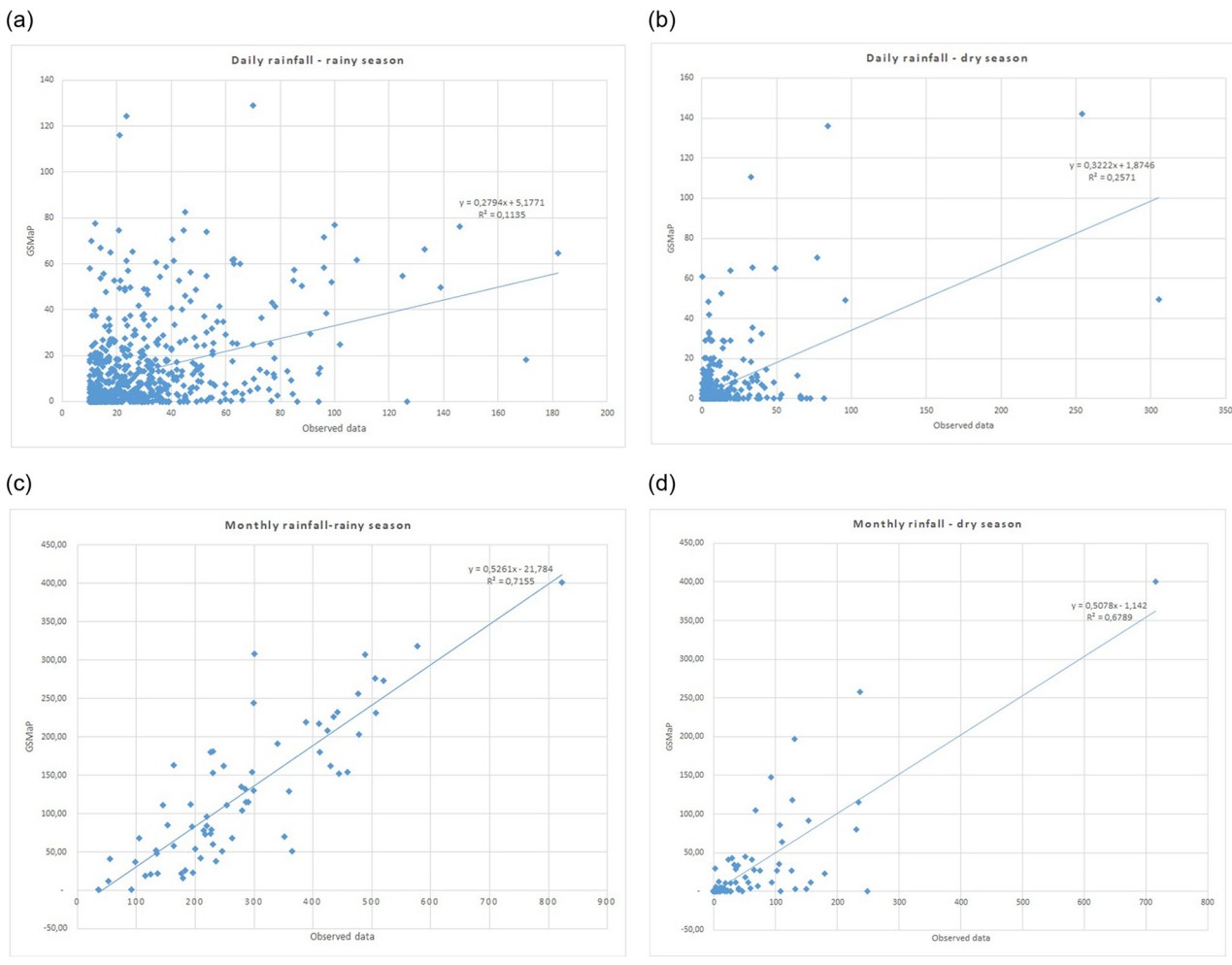

**Fig 7. Correlation analysis of satellite rainfall data and observed data.** (a). Daily rainfall rainy season, (b). Daily rainfall dry season, (c). Monthly rainfall rainy season, (d). Monthly rainfall dry season.

simple field study discovered 13 locations with natural valleys on stream orders 1 and 2. Fig 10 depicts sites with natural valleys.

## Discussion

### Satellite climate data reliability

Due to improvements in satellite rainfall products, it is now possible to use satellite rainfall data to fill the gap left by the temporary and spatially limited nature of observed rainfall data [43–45]. There is only one rainfall station at the site of this study. Consequently, it is crucial to employ satellite data. Prior to the analysis, satellite data must be verified to observed data.

**Table 5. Maximum monthly rainfall (mm) by return period.**

| Return Period (year) | 2 | 5 | 10 | 20 | 50 | 100 |
|---|---|---|---|---|---|---|
| Rain (mm)-upstream | 320 | 388 | 433 | 476 | 532 | 573 |
| Rain (mm)-downstream | 336 | 400 | 442 | 482 | 534 | 574 |

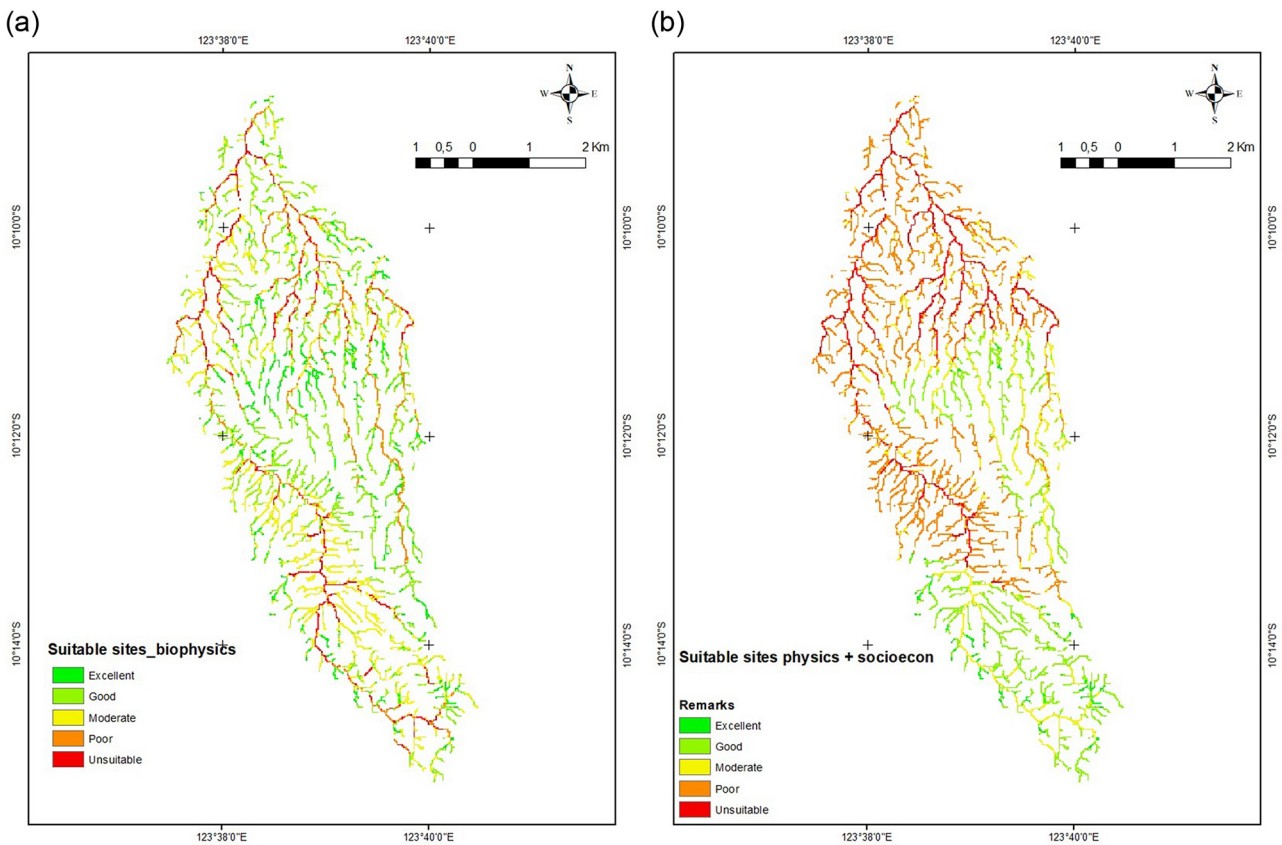

**Fig 8. GIS based suitable sites.** (a) biophysical aspects, (b) biophysical and socio-economic aspects.

According to statistical parameter analysis, the daily rainfall correlation coefficient is weak during the summer and moderate during the rainy season. Similar findings are made on Bali, where the daily rainfall correlation coefficient for GSMaP is low [46]. However, this finding contrasts with that of a Chinese study [21], which discovered a high correlation between GSMaP daily precipitation data and the observational data. This demonstrates that there is still uncertainty over the applicability of satellite data, particularly in mountainous and arid regions [47, 48].

Correlation values for monthly rainfall throughout the summer and wet season are very strong and strong, respectively. This work supports previous research in an arid mountainous basin, the Qaraqash River basin, that the monthly satellite precipitation correlation coefficient is quite high [47]. A study in Bali island reported similar findings suggesting that the higher the temporal scale, the stronger the GSMaP correlation coefficient [46].

## GIS-based MCA suitable sites

GIS applications by processing different criteria (GIS-based MCA) can be used to locate RWH and can be applied to a wider watershed, according to research conducted in India on a 210 km2 watershed [7]. In Iraq, research conducted over a larger region (16.72 km2) revealed that the use of GIS-based MCA is acceptable [24]. The results of this study, which used a smaller watershed in the GIS-based MCA processing, are strongly reliable. This demonstrates the GIS application's robustness in the RWH sites identification.

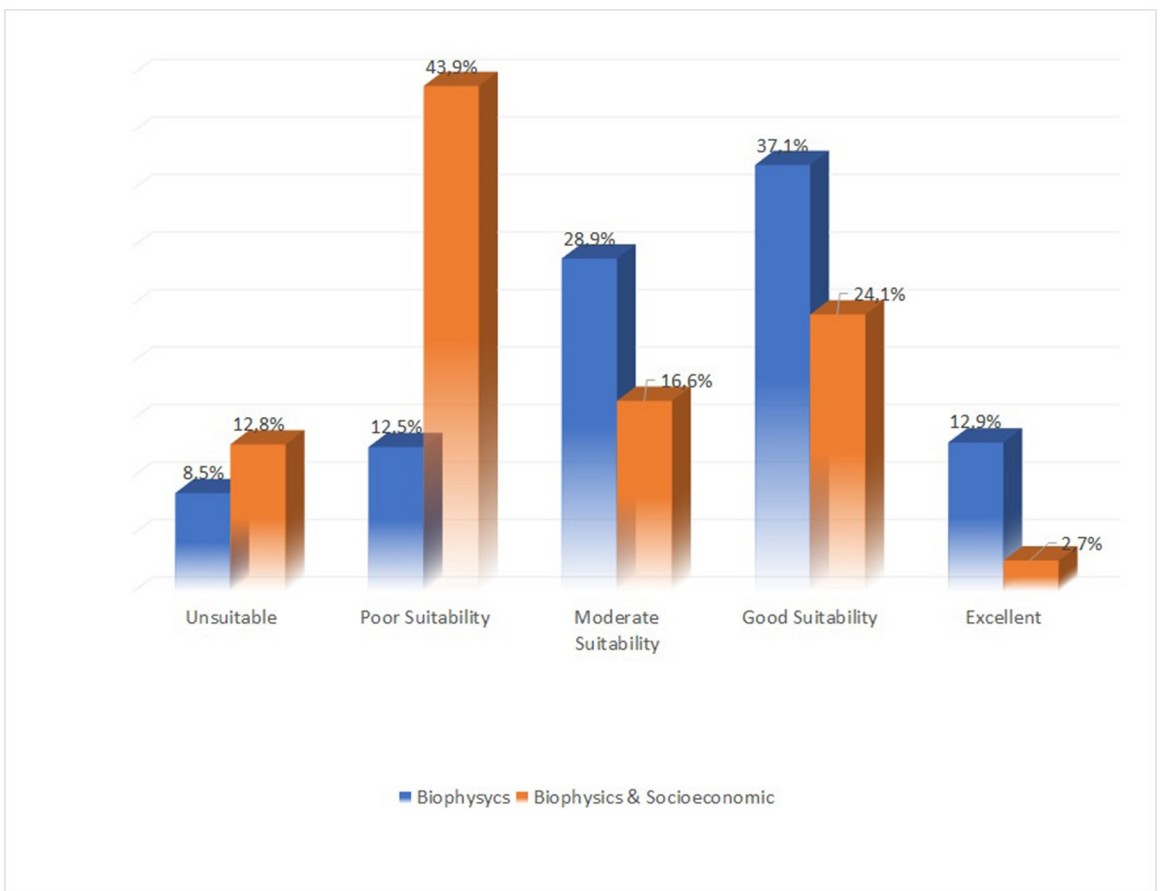

**Fig 9. Percentage of area covered by the different *embung* sites suitability.**

Desktop analysis using GIS tools is very effective, however the results should be evaluated in light of the geophysical characteristics of the area and the needs of the local community [3]. According to this study, half of the first and second order streams are suitable or very suitable for ponds when the biophysical factors were taken into account. The rate drops to 27% when socioeconomic factors are considered. This demonstrates how fundamental it is to take socioeconomic factors into account while planning the infrastructure for water resources [18]. As the community is the direct beneficiary of the pond, it is crucial to discuss the findings with them.

## Conclusions

The project aims to develop a model for identifying the *embung* location utilizing GIS-based MCA techniques and satellite rainfall data, and to validate the model using simple field observations. It suggests that the use of GSMaP rainfall data from satellites plays a crucial practical role, particularly in hydrological analysis in areas with limited observational rainfall data. Daily rainfall is unreliable, though, as actual rainfall is frequently underestimated by satellite data. In order to be used in hydrological analysis, the monthly satellite rainfall data has a high and extremely strong correlation. The use of the GIS-based MCA approach demonstrates that 13% of the overall stream system are not suitable for *embung*, whereas locations that are poorly fit and moderately fit make up 44% and 17%, respectively. Sites with suitable and extremely suitable for ponds

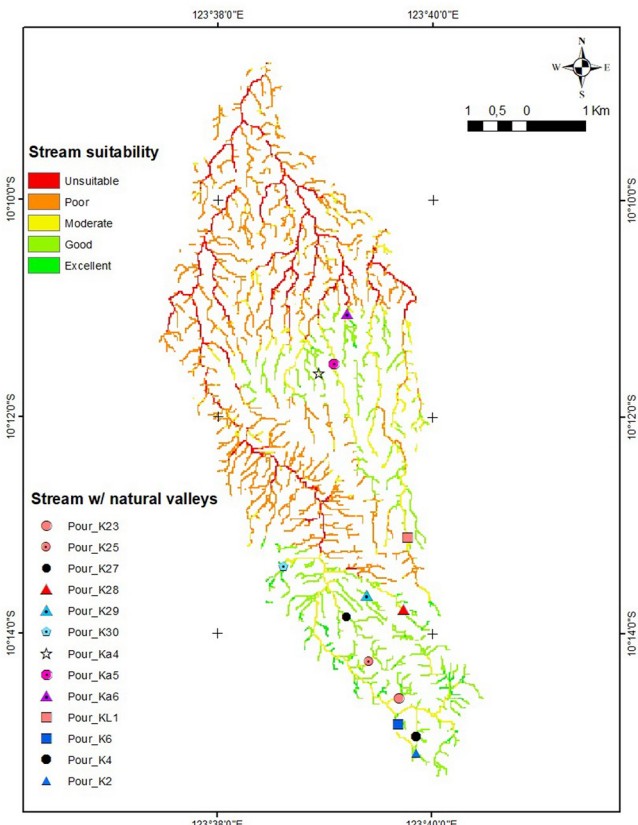

**Fig 10. Observation and GIS based suitable sites for *embung*.**

make up 27%. This model is very useful in determining several alternative pond's locations including the size of the catchment area and input discharge. As a result of observation, it has been confirmed that there is no natural flow in several stream orders 1 and 2 with small catchment areas. It was found that only 13 natural valleys were discovered from observations. These areas are likely candidates for ponds with minimal need for excavation. Because of their socioeconomic and environmental difficulties, other areas are only partially eligible.

The combination of geospatial data, GIS, a multi-criteria analysis, and a field survey proved effective for the RWH site selection in a semi-arid region with limited data, especially on the first and second order streams. It is recommended that decision-makers and managers of water resources employ this quick and cheap technique since it offers reasonable alternative sites for RWH.

## Supporting information

**S1 Data. GSMaP rainfall data.**
(XLSX)

**S1 File. Raster file.**
(ZIP)

**S2 File. Translation of BIG letter.**
(DOCX)

**S3 File. License from BIG.**
(PDF)

## Acknowledgments

The authors would like to thank all parties for their supports.

## Author Contributions

**Conceptualization:** Yulius Patrisius Kau Suni, Joko Sujono,  Istiarto.

**Data curation:** Yulius Patrisius Kau Suni.

**Formal analysis:** Yulius Patrisius Kau Suni, Joko Sujono,  Istiarto.

**Investigation:** Yulius Patrisius Kau Suni.

**Methodology:** Yulius Patrisius Kau Suni, Joko Sujono.

**Project administration:** Yulius Patrisius Kau Suni.

**Resources:** Yulius Patrisius Kau Suni.

**Software:** Yulius Patrisius Kau Suni,  Istiarto.

**Supervision:** Joko Sujono,  Istiarto.

**Validation:** Yulius Patrisius Kau Suni, Joko Sujono,  Istiarto.

**Visualization:** Yulius Patrisius Kau Suni.

**Writing – original draft:** Yulius Patrisius Kau Suni.

**Writing – review & editing:** Yulius Patrisius Kau Suni.

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
