## [Decision Letter · Decision Letter 0]

5 Feb 2023

PONE-D-23-00571Identifying potential sites for rainwater harvesting ponds (embung) in Indonesia’s semi-arid region using GIS-based MCA techniques and satellite rainfall dataPLOS ONE

Dear Dr. Suni,

Thank you for submitting your manuscript to PLOS ONE. After careful consideration, we feel that it has merit but does not fully meet PLOS ONE’s publication criteria as it currently stands. Therefore, we invite you to submit a revised version of the manuscript that addresses the points raised during the review process.

We look forward to receiving your revised manuscript.

Kind regards,

Zaher Mundher Yaseen

Academic Editor

PLOS ONE

Journal Requirements:

3. We note that Figures 1 to 5 in your submission contain [map/satellite] images which may be copyrighted. All PLOS content is published under the Creative Commons Attribution License (CC BY 4.0), which means that the manuscript, images, and Supporting Information files will be freely available online, and any third party is permitted to access, download, copy, distribute, and use these materials in any way, even commercially, with proper attribution. For these reasons, we cannot publish previously copyrighted maps or satellite images created using proprietary data, such as Google software (Google Maps, Street View, and Earth). For more information, see our copyright guidelines: http://journals.plos.org/plosone/s/licenses-and-copyright.

a. You may seek permission from the original copyright holder of Figures 1 to 5 to publish the content specifically under the CC BY 4.0 license.  

Additional Editor Comments:

The suggested references by the reviewers to be neglected.

Reviewers' comments:

Reviewer's Responses to Questions

**Comments to the Author**

1. Is the manuscript technically sound, and do the data support the conclusions?

Reviewer #1: Partly

Reviewer #2: Yes

2. Has the statistical analysis been performed appropriately and rigorously? 

Reviewer #1: Yes

Reviewer #2: Yes

3. Have the authors made all data underlying the findings in their manuscript fully available?

Reviewer #1: Yes

Reviewer #2: Yes

4. Is the manuscript presented in an intelligible fashion and written in standard English?

Reviewer #1: Yes

Reviewer #2: Yes

5. Review Comments to the Author

Reviewer #1: The manuscript aims to develop a model for identifying potential embung locations in 1st and 2nd stream orders based on satellite data (rainfall, topography, soil conditions, and land use) using a combination of GIS-based MCA techniques in a semi-arid area of Indonesia, Liliba watershed, Timor. This study employs a geo-information system (GIS) based multi-criteria analysis (MCA) approach and satellite rainfall data, Global Satellite Mapping of Precipitation (GSMaP) to determine the suitable locations for the ponds. The proposed approach is not completely clear, some steps have to be better explained:

1- The abstract is generally well written but some modifications are suggested. Authors are requested to add the information how the criteria are selected?

2- The future recommendations should also be provided in abstract section

3- Line 67 Please write Geographic Information System (GIS).

4- The authors did not mention about the selections of criteria.

5- Map of the study area is not clear. Authors are requested to enhance the resolution and size of the figure.

6- It would be better to add the rainfall and temperature maps in the manuscript, so that the readers clearly understand the distribution pattern of rainfall, temperature, runoff etc.

7- Line 79 Support this sentence with proper references such as

doi:10.1088/1757-899X/881/1/012170; https://doi.org/10.1155/2021/5580286

8- Line 87 -88 please write the longitudes and latitudes in correct forms such as 123⁰37′23.3″ .

9- Data collection are not well captured hence suffering from many shortcomings.

10- Please add more information about Land cover and land use, how the authors is conducted, which method has been used, the name of classifier, classification parameters, accuracy of classification---etc.

11- Table 1. Correct Area km2.

12- Proper justification on the selection of the binary technique and the Weighted Linear Combination (WLC) in the Geographic information system (GIS), should be reported with proper referencing.

13- Proper justification on the selection of the AHP should be reported with proper referencing.

14- Line 227-235 the rainfall runoff relationship is discussed in many references please, add more references such as https://doi.org/10.18280/ijdne.150511;

https://doi.org/10.1007/s12518-021-003;
https://doi.org/10.18280/ijdne.150318;

15- Line 287 please correct Area km2

16- About the use of the WLC. First, I ask the authors to specify that the operation to refer all criteria to the same scale must be referred as ‘normalisation’. The authors must also specify that they used a raster model.

17- Are there any sensitivity analyses conducted for the selected criteria?

18- Authors are requested to add future recommendations for the study area.

Reviewer #2: 1. Provide reference/citaions for the suitability/unsuitability of soils for reservoir (Table 2. Soil types in Liliba watershed).

2. Provide reference for the suitability/unsuitability of land covers for reservoir (Table 1. Land cover and land use statistical data).

3. Improve citations overall the paper.

4. Include high quality figures.

5. include appropriate titles for all figures.

6. make border scale for all figures.

6. PLOS authors have the option to publish the peer review history of their article (what does this mean?). If published, this will include your full peer review and any attached files.

Reviewer #1: **Yes: **Khamis N. Sayl

Reviewer #2: No

---

## [Author Response · Author response to Decision Letter 0]

17 Apr 2023

All comments from reviewers and academic editor were responded in the file attached.

---

## [Decision Letter · Decision Letter 1]

9 May 2023

Identifying potential sites for rainwater harvesting ponds (embung) in Indonesia’s semi-arid region using GIS-based MCA techniques and satellite rainfall data

PONE-D-23-00571R1

Dear Dr. Suni,

We’re pleased to inform you that your manuscript has been judged scientifically suitable for publication and will be formally accepted for publication once it meets all outstanding technical requirements.

Kind regards,

Zaher Mundher Yaseen

Academic Editor

PLOS ONE

Additional Editor Comments (optional):

Reviewers' comments:

Reviewer's Responses to Questions

**Comments to the Author**

1. If the authors have adequately addressed your comments raised in a previous round of review and you feel that this manuscript is now acceptable for publication, you may indicate that here to bypass the “Comments to the Author” section, enter your conflict of interest statement in the “Confidential to Editor” section, and submit your "Accept" recommendation.

Reviewer #1: All comments have been addressed

Reviewer #2: All comments have been addressed

2. Is the manuscript technically sound, and do the data support the conclusions?

Reviewer #1: Yes

Reviewer #2: Yes

3. Has the statistical analysis been performed appropriately and rigorously? 

Reviewer #1: Yes

Reviewer #2: Yes

4. Have the authors made all data underlying the findings in their manuscript fully available?

Reviewer #1: Yes

Reviewer #2: Yes

5. Is the manuscript presented in an intelligible fashion and written in standard English?

Reviewer #1: Yes

Reviewer #2: Yes

6. Review Comments to the Author

Reviewer #1: The authors have been addressed all my comments and suggestions therefor I think the manuscript based on last modification is ready for publication.

Reviewer #2: (No Response)

7. PLOS authors have the option to publish the peer review history of their article (what does this mean?). If published, this will include your full peer review and any attached files.

Reviewer #1: No

Reviewer #2: **Yes: **Ziaul Haq Doost

---

## [Editor Report · Acceptance letter]

16 May 2023

PONE-D-23-00571R1 

Identifying potential sites for rainwater harvesting ponds (*embung*) in Indonesia’s semi-arid region using GIS-based MCA techniques and satellite rainfall data 

Dear Dr. Suni:

I'm pleased to inform you that your manuscript has been deemed suitable for publication in PLOS ONE. Congratulations! Your manuscript is now with our production department. 

Kind regards, 

on behalf of

Dr. Zaher Mundher Yaseen 

Academic Editor

PLOS ONE